# Knowledge Graph Lifecycle: Building and maintaining Knowledge Graphs

Umutcan Şimşek[1], Kevin Angele[1] Elias Kärle[1,2], Juliette Opdenplatz[1], Dennis Sommer[1], Jürgen Umbrich[2], and Dieter Fensel[1]

[1] University of Innsbruck, Technikerstrasse 21a 6020 Innsbruck, Austria
`firstname.lastname@sti2.at`
[2] Onlim GmbH `firstname.lastname@onlim.com`

**Abstract.** Knowledge Graphs can integrate a large amount of data from heterogeneous sources. They can be an important resource for various applications; however, they are only useful if they satisfy the requirements of those applications in terms of quality. In this in use experience paper, we present our approach and tools for supporting the Knowledge Graph Lifecycle that starts with creation and hosting and continues with the curation and deployment. The curation process enables the mainteinance of a Knowledge Graph, especially in terms of correctness and completeness. We provide process models and evaluation of developed tools with Knowledge Graphs in tourism domain. We discuss the lessons learned from implementing such an approach in an open and commercial setting and conclude with a summary and a description of currently ongoing use cases.

**Keywords:** knowledge graphs · knowledge graph lifecycle · knowledge curation · knowledge creation

## 1 Introduction

Knowledge Graphs integrate data from heterogeneous sources and can rapidly become quite large [5, 7]. They can be a useful resource to power many applications in different domains. To enable these applications, the scalable construction and maintenance of Knowledge Graphs are crucial. The lifecycle of a Knowledge Graph comes with two main challenges (1) how to integrate heterogeneous sources in a Knowledge Graph in a scalable manner (2) how to make them a high-quality resource (e.g., semantically and syntactically correct, no duplicate instances) given the applications in hand.

In this in-use experience paper, we explain the methodology and tools to support the Knowledge Graph Lifecycle, which consists of the creation, hosting, curation, and deployment of a Knowledge Graph. The Knowledge Graphs built with this approach are deployed in an open as well as a commercial setting in the tourism domain to support conversational agents. We discuss the lessons learned while implementing our approach.

In the remainder of the paper, we first present our methodology and tools for the Knowledge Graph lifecycle (Section 2). Then, we discuss the lessons learned from developing and implementing our methodology (Section 3). Finally, we provide concluding remarks and indicators for the future work (Section 4)[3].

## 2   Knowledge Graph Lifecycle

There are already proposed methodologies for iterative construction of Knowledge Graphs from various sources (a recent one is described in [12]), but construction is only the one side of the coin. On the one hand, it must be built from various heterogenous sources, on the other hand, it must be turned into a high-quality resource that satisfies the requirements of the use case and applications in hand [14]. Figure 1 shows these processes and the tools developed to support them.

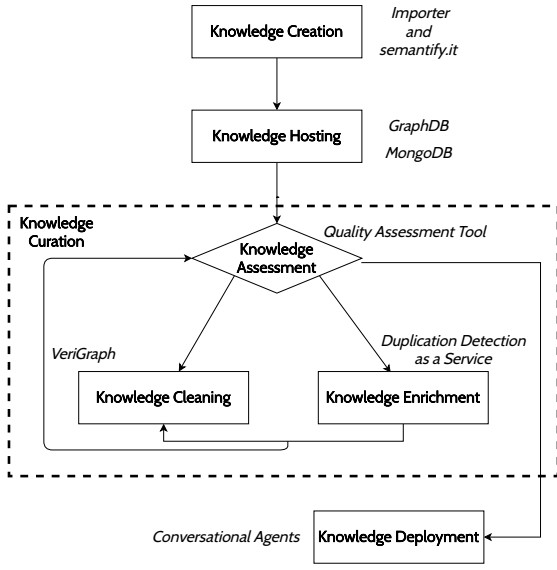

Fig. 1: The Knowledge Graph Lifecycle (adapted from [5]). The italic labels represent the tools developed/used for each process

The lifecycle starts with the creation process that deals with the creation of semantic data from heterogeneous sources. This semantic data is then hosted in

---

[3] Note that we do not have a dedicated related work section. Naturally, we benefited from a plethora of research work while developing our approach. Unfortunately, we cannot mention them all in this experience paper due to space restrictions, but only the ones that are directly related to each process. A comprehensive review can be found in [5].

various stores (depending on the deployment target) and curated. The curation process first assesses the quality and based on the assessment result trigger cleaning and enrichment processes to improve its correctness and completeness. This process continues iteratively to constiute the lifecycle. In the remainder of the section, the process model, tool, and evaluation for each step in the lifecycle are presented.

### 2.1   Knowledge Creation

Knowledge creation[4], as defined in [5], describes "extracting information from different sources, structuring it, and creating useful knowledge". For the creation process, we use schema.org vocabulary as schema. The vocabulary covers many domains in a shallow way and is a de facto industrial standard for semantic annotations on the web. We create domain-specific patterns of schema.org to facilitate knowledge creation. These patterns are extended subsets of schema.org. These subsets define local properties on types from schema.org its extensions and constraints on those properties (see also Section 2.3).

   We focus on two different ways of creating knowledge: manual and via mappings. A free-to-use deployment of both knowledge creation tools can be found in the semantify.it platform [8][5].

**Manual**  The manual knowledge creation refers to creating RDF graphs via a form-based interface. The form-based interface is dynamically generated based on a domain-specific pattern [16], which is a SHACL node shape (see also 2.3).

   The manual creation process is typically used for low-volume, unstructured, and low-volatility content and data. For example, annotation of web pages that provide relatively static information. To make this process a bit easier, we provided tool support as part of the semantify.it platform . The platform offers a generic annotation editor that provides a form-based interface based on domain-specific patterns for the annotation of content and data.

**Mapping**  The majority of the knowledge used for building Knowledge Graphs in our case is created through declarative mappings from semi-structured data sources. In the context of our work, the data to be integrated into the knowledge graph was obtained from different service providers in different formats, typically JSON or XML. The mapping was then defined from those sources to schema.org.

   The process model for knowledge creation via mappings is the following: We first collect raw data from different service providers via web services. Each object retrieved is mapped to schema.org to create an instance of a schema.org type with its property values assertions. The generated RDF data is then enriched with provenance information based on PROV-O[6].

---

[4] It may be also called knowledge acquisition
[5] https://semantify.it - registration and login required.
[6] https://www.w3.org/TR/prov-o

We implemented this process in the Importer tool. The tool allows registration of new sources including all the access information and their RML[3] mapping files. The timing and frequency of the mappings can be specified with cron strings. The Importer has Apache NiFi[7] in its core to manage the entire dataflow from accessing raw data to storing in a triple store. Apache NiFi is a dataflow management tool that offers load balancing, buffering, and guaranteed delivery. The actual mapping is executed via an external RocketRML[17][8] instance, a scalable RML mapper implemented with NodeJS. RocketRML currently supports JSON, XML, and CSV formats and adopts optimization techniques like JOIN path memoization for high-performance. It supports various JSON-Path and XPath implementations allowing users to access extra features like backward traversal in a JSON file with JSON-Path Plus[9]. It also supports function mappings which are frequently used for transforming property values and distinguishing between different subtypes of schema.org type during the mapping (e.g., different types of events can be dynamically mapped with a single mapping).

The RML mapper used in the Importer, RocketRML, can map 25K triples per second in average. However, the overhead caused by sending queries to the GraphDB instance over HTTP has a negative impact on the overall import process. A more detailed explanation of knowledge creation via mappings and a detailed evaluation of the importer tool can be found in [18].

## 2.2   Knowledge Hosting

We host the created knowledge on two different stores namely a MongoDB instance as a document store, and a GraphDB Enterprise instance. With the creation approach above we create various Knowledge Graphs, in fact, each user of the semantify.it platform can create its Knowledge Graphs via mappings. They can additionally publish the created instances of schema.org types on their web pages for purposes like semantic search engine optimization. The web annotations are stored in a MongoDB instance as JSON-LD documents to ease the access to individual annotations, as it is challenging to retrieve all properties describing an instance directly from a triplestore via SPARQL.

We create both commercial and open knowledge graphs with the aforementioned approach. A notable example is the Tyrolean Tourism Knowledge Graph[10], which contains more than 12B statements. It is populated with data from 11 different sources (mainly Destination Management Organizations from different regions in Tyrol) and updated daily. The knowledge coming from different sources is organized in named graphs. The named graphs imported from the same source on different time points are linked with each other via the provenance information, which allows applications like time series analysis on frequently changing data (e.g., accommodation prices, weather measurements).

---

[7] https://nifi.apache.org
[8] https://github.com/semantifyit/RocketRML
[9] https://www.npmjs.com/package/jsonpath-plus
[10] http://tirol.kg

### 2.3   Knowledge Curation

Knowledge Curation is a process for assessing and improving a Knowledge Graph in various dimensions, especially correctness and completeness (see also "knowledge refinement" [11], with a narrower set of tasks). In this section, we explain the processes comprising Knowledge Curation and the tasks on which we focused in the scope of our work.

**Knowledge Assessment**  Knowledge Assessment is the process of assessing the quality of Knowledge Graphs. The quality is measured on multiple dimensions using different metrics. In our case, the dimensions and metrics have been selected from the (linked) data quality literature (e.g., [2], [4], [15]). After eliminating some redundancies, we ended up with 20 dimensions (e.g., Accessibility, Correctness, Completeness, Consistent Representation) with a total of 41 metrics (e.g., Accessibility dimensions has a metric for whether a Knowledge Graph offers a SPARQL endpoint).

In many application scenarios, different quality dimensions have a different degree of importance for different applications and domains. Similarly, different metrics may have different importance for the calculation of the quality score for a dimension. For example, the Timeliness dimension may not be very important in a domain that has predominantly static data. Therefore, we allow quality dimensions and metrics to be weighted for different domains (i.e., for instances of different types). The sum of the weights for metrics for each dimension and the sum of the weights for each dimension amounts to 1. The process model for Knowledge Assessment is shown in Figure 2.

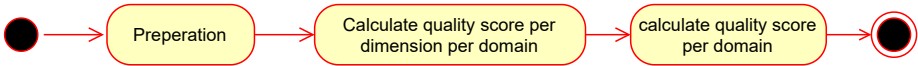

Fig. 2: UML Activity Diagram for the Knowledge Assessment process

*Preperation* is the activity that deals with various configuration steps that are necessary for the assessment of a Knowledge Graph. This includes the definition of weights for different metrics and dimensions for different domains, as well as the dimension-specific configuration (e.g., where to find RDF dumps for Accessibility dimension, which property values should the instances contain for each domain to calculate completeness metrics, specification of integrity constraints).

*Calculating quality score per dimension per domain* step produces a score between 0 and 1 for each dimension based on adding up its weighted metric scores. The score of a metric in a dimension is multiplied by the weight defined at the previous activity to obtain a weighted metric score. The extent of automation for the calculation depends on the nature of the Knowledge Graph. For example,

if the Knowledge Graph has its license in a machine-readable format, then the metric scores involving licenses can be calculated in an automated fashion. Moreover, some of the metrics, especially regarding semantic correctness of the data simply cannot be fully automated with the intrinsic knowledge in the Knowledge Graph. To check whether the address of a place is correct, the assessment process needs external knowledge, either a human observer or an authoritative source. This may not be scalable on a very large Knowledge Graph, therefore sampling may be needed for such an assessment.

*Calculating weighted aggregate quality score per domain* step takes the score of each dimension and creates an aggregated score based on the weights defined for each dimension for each domain. The score is between 0 and 1.

The described process model has been implemented in a Knowledge Assessment Framework called QAT[11] (**Q**uality **A**ssessment **T**ool). QAT is implemented as Software as a Service (SaaS) that periodically[12] fetches the information from the configured data sources automatically, whenever possible. Other metrics can be assessed by a user and assigned to the corresponding metrics. Similarly, a user can define customized weights to the metrics and dimensions and can access the result of the overall score either via an API[13] or a user interface. The evaluation of the tool is still ongoing.

**Knowledge Cleaning** Knowledge Cleaning is a process that aims to improve the correctness of a Knowledge Graph. It consists of (a) error detection, the task for identifying the erroneous type and property value assertions (b) error correction, fixing the identified statements. We focused on the former, particularly the verification task where the Knowledge Graph is checked against a specification such as integrity constraints[14].

Our approach is based on verifying the instances in a Knowledge Graph against the domain-specific patterns of schema.org. These patterns are expressed with a subset of SHACL [15].

For detecting errors in a Knowledge Graph, we conceptualized and developed a verifier that checks whether a particular subset of a Knowledge Graph fits the domain-specific pattern. Figure 3 shows the process model.

The first step loads a domain-specific pattern that comprises the shapes graph for verification. Then the verification process is split into two lines: The first line of verification retrieves the URIs of the instances that match the target specification and adds them to a verification queue. Then, for each URI in the queue, a data graph is retrieved and verified against the domain-specific pattern.

---

[11] https://qat.semantify.it/

[12] Assessment on demand is part of the future work.

[13] https://qat.semantify.it/datasources - API that returns the assessment values for the implemented domains and data sources using default weights.

[14] The validation tasks complements verification by checking the Knowledge Graph against the "real-world". This task has not been covered in our work yet.

[15] Details of the subset can be found in [16]

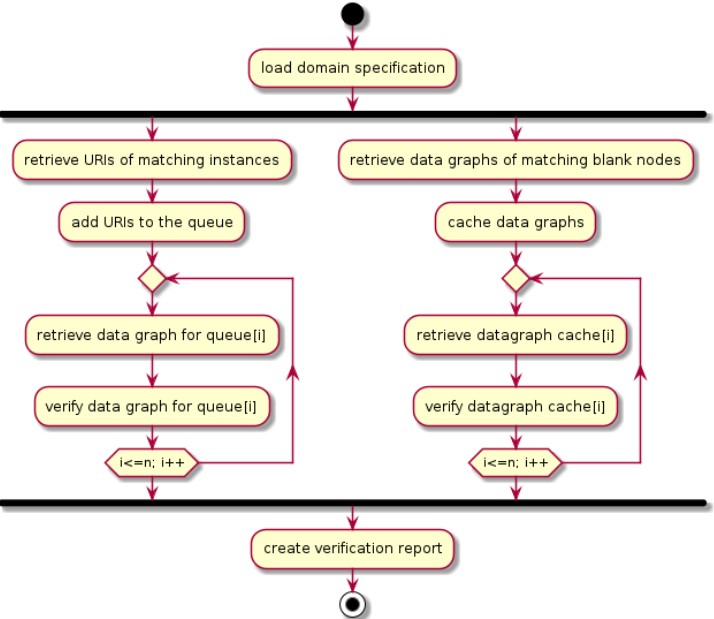

Fig. 3: UML Activity Diagram for error detection process model

In parallel, the verification process retrieves all blank nodes that match the target description and their data graphs and stores them in a cache. Then, each data graph in the cache is verified against the loaded domain-specific pattern. The results of both lines of verification are then compiled in a verification report. Note that the data graph in both lines corresponds to the subgraph built by following the all outgoing edges of a focus node recursively until no new node can be added to the data graph (e.g., all nodes to be expanded are literals)[16].

We implemented our error detection approach in the VeriGraph tool[17]. The tool has been implemented in Javascript and available with an open license. It can be configured to run on any Knowledge Graph that provides a SPARQL endpoint. In our experience with many SHACL verifiers, we realized there are generally two main issues in practice: (1) operating in-memory, which causes insufficient memory problems with large data graphs (2) SPARQL endpoints are not always reliable for frequent queries that return high-volume results. The first issue we address with the caching mechanism. The data graphs are cached on the disk and only loaded to the memory when they are needed for verification.

---

[16] Due to the limitations of SPARQL, such a recursive traversal is tricky. There is a way to do this in a single query by using `?s (:|!:)* ?o` graph pattern, but admittedly it is a bit *hacky*. One can also rely on DESCRIBE queries if the triplestore implements it appropriately.

[17] https://github.com/semantifyit/VeriGraph

The second issue is addressed by both indexing and the caching mechanism. Indexing the URIs and querying their data graphs one-by-one reduces the size of the data graph returned by a single query[18]. Each constraint component defined by the property shapes is checked by graph-traversal in the memory. This reduces the number of SPARQL queries running against an endpoint for verification. Additional to the typical SHACL verification report, the VeriGraph tool provides metadata about the verification process (e.g., duration, number of violations found).

We evaluated the VeriGraph tool on several Knowledge Graphs with an increasing number of triples, starting from 100K up to 1B[19]. Each Knowledge Graph contains instances of types like Event, Hotel, HotelRoom, Person, and Product. The instances were verified against a set of constraints with different target specifications[20]. The evaluation has been conducted on a server with an Intel Core i9-9900K Octa-Core 3.60GHz processor, 64GB RAM, and 2TB SSD. We compared our tool with AllegroGraph, RDFUnit, ShExJava, Stardog ICV, and TopBraid. Figure 4 shows the verification time in relation to the size of Knowledge Graph size. VeriGraph stands out as the size grows, as it is the only one that can finish verification on a Knowledge Graph with 1B triples. In smaller Knowledge Graphs, VeriGraph is only better than RDFUnit. This can be explained by other tools either working completely in-memory (ShExJava and TopBraid) or on their triplestores natively (AllegroGraph, Stardog). For RDFUnit and VeriGraph connecting to generic SPARQL endpoints of triplestores create an overhead.

**Knowledge Enrichment**  Knowledge Enrichment is a process that aims to improve the completeness of a Knowledge Graph [5]. The completeness is enhanced by identifying and adding missing instance, property value, and equality assertions. In our work, we focused on the duplicate detection task, to find the equality assertions between instances within or across Knowledge Graphs[21].

---

[18] Such indexing is not possible with blank nodes as they are locally-scoped and cannot be directly addressed in a SPARQL query. However, we have a "retail mode" that can be configured for using the internal identifiers given to a blank node by a triple store. These identifiers are naturally vendor-specific, however, improve the verification performance significantly.

[19] http://dataset.sti2.at/datasets/. Large datasets have been created based on Tyrolean Tourism Knowledge Graph.

[20] https://github.com/semantifyit/VeriGraph/blob/master/constraints/constraints.ttl

[21] We take schema.org as the golden standard and do not focus on the alignment of TBox. Heterogenous schemas from different Knowledge Graphs are mapped to schema.org via declarative mappings such as RML mappings.

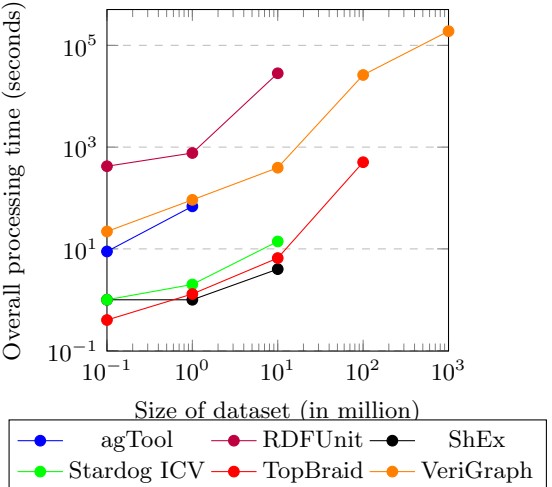

Fig. 4: Verification time - Knowledge Graph size plot for different tools [1]

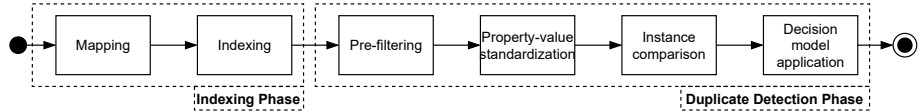

Fig. 5: A short depiction of the duplicate detection process.

We developed a highly configurable service-oriented approach to the duplicate detection problem that allows linking duplicate instances in a Knowledge Graph or from external Knowledge Graphs [10]. The process model is shown in Figure 5. The whole process is divided into two main phases: (1) the indexing phase where the knowledge sources are mapped to a common format if needed and indexed into an internal Elasticsearch[22] instance, and (2) the duplicate detection phase which is divided into four steps: The first step is the pre-filtering step that determines candidate duplicates of two previously indexed knowledge sources. This step selects a set of candidate instances to which the duplicate detection process will pay attention[23]. In the second step properties are normalized such that two instances are easier to compare (e.g., via regular expressions over string values, mathematical operations to normalize units of certain numerical values). The third step executes the actual detailed comparison between the

---

[22] https://www.elastic.co/elasticsearch/

[23] The pre-filtering works based on the more_like_this queries of elasticsearch. The "likeness" is calculated with TF-IDF. Putting a high threshold for the number of matching terms may harm the recall of the overall approach. See https://www.elastic.co/guide/en/elasticsearch/reference/current/query-dsl-mlt-query.html .

candidate duplicates which results in a similarity score for a candidate duplicate. Here several different similarity metrics are used for different types of property values (e.g., Jaccard, Levenshtein for string similarity; Euclidean distance for geocoordinates). Finally, the fourth step applies a decision model in which the similarities from the third step are utilized to classify the suspected duplicates as either duplicates or non-duplicates. The output of the duplicate detection phase can then simply be translated into *schema:sameAs* statements.

Both indexing and duplicate detection steps need to be configured. For example, the threshold for pre-filtering must be specified, and the normalization functions together with similarity metrics must be specified for each property (or a combination of them) relevant to the duplicate detection process. The configuration task requires a lot of a priori knowledge about the application and the data. To tackle this challenge, our approach offers supervised methods to learn configurations from a sample set of linked instances (i.e., a golden standard). We use various algorithms such as time-constrained brute force, hill climbing, localized brute force, and genetic algorithms with random mutations to learn the configuration parameters for different steps. The configuration learning can be optimized towards improving recall, precision, or F-score.

We implemented our approach in a tool called Duplicate Detection as a Service (DDaaS). DDaaS was developed as a general-purpose duplicate detection framework that consists of easily exchangeable and extensible components. The tool consists of multiple services for different tasks. These services are orchestrated via REST. In future iterations, the services are going to be more loosely coupled and orchestrated using an Apache Kafka instance.

To evaluate the approach, we compared it with three other tools, Duke [6], LIMES [9], Silk [13] all of which also influenced the development of DDaaS. We compared the tools over two datasets (Restaurants[24] and SPIMBENCH[25]). The results are displayed in Table 1.

| Restaurants | | | | | SPIMBENCH | | | |
|---|---|---|---|---|---|---|---|---|
| Tool | F1-Score | Precision | Recall | | Tool | F1-Score | Precision | Recall |
| DDaaS | 0.76 | 1.00 | 0.61 | | DDaaS | 0.85 | 0.98 | 0.76 |
| Duke | 0.77 | 1.00 | 0.62 | | Duke | 0.09 | 0.05 | 0.75 |
| LIMES | 0.80 | 0.86 | 0.74 | | LIMES | 0.72 | 0.88 | 0.61 |
| Silk | 0.40 | 0.79 | 0.27 | | Silk | 0.62 | 0.77 | 0.53 |

Table 1: Duplicate detection comparison of DDaaS, Duke, LIMES, and Silk

We ran every tool the most automated way possible and results indicate that DDaaS is at least on par with the other tools. The SPIMBENCH results are particularly interesting here, while the results for the Restaurants dataset are more balanced. The main differences between these datasets lie in the number of

---

[24] https://www.cs.utexas.edu/users/ml/riddle/
[25] https://project-hobbit.eu/challenges/om2020/

properties and their completeness. While the Restaurant dataset is a perfect *toy* dataset, the instances in the SPIMBENCH dataset are very sparse with regards to property values on many instances. Duke's purely genetic approach to learning a configuration for this particular dataset seems to be flawed as it will keep retrying to use properties that are not even available for most instances. DDaaS' approach to learning a configuration is a composition of different approaches which includes a genetic approach but also less randomized approaches. We call this composition the learning strategy. Since every single element of this strategy (i.e., configurable pieces) can be configured to aim to optimize one of the three measures (precision/recall/F1), it achieved a good performance even on such an incomplete dataset. Further evaluation on Tyrolean Tourism Knowledge Graph is ongoing.

### 2.4   Knowledge Deployment

"The proof of the pudding is in the eating." A Knowledge Graph is only as valuable as the applications it enables. Therefore, the knowledge deployment task deals with the applications that are powered by a Knowledge Graph. Alongside open Knowledge Graphs like Tyrolean Tourism Knowledge Graph, the Knowledge Graphs developed with the presented approach are deployed commercially by Onlim GmbH. Onlim uses their Knowledge Graph to power their conversational agents in different domains, most notably tourism. They provide about 20 conversational agents for customers in the tourism domain. These agents are typically goal-oriented dialog systems (GDS) that help users to achieve their goals via conversations. In the case of Onlim, a Knowledge Graph powers a GDS in two different ways:

– Providing entities for annotating user utterances to train Natural Language Understanding (NLU) models.
– Serving as a knowledge source to provide the knowledge needed for a task at hand

Onlim uses state-of-the-art GDS development frameworks such as DialogFlow[26] and RASA[27] to streamline the conversational aspects of a dialog system such as NLU, dialog management and NLG. Such frameworks work with intents, structures that represent the user goals a GDS supports. The frameworks use supervised machine learning to classify utterances to intents. They use Knowledge Graphs to create annotated utterances for each intent to help the machine learning models classify incoming utterances to the correct intents. An intent is then mapped to a SPARQL query and the user's question is answered based on the data provided by the Knowledge Graph (e.g., accommodation, events, infrastructure). The lifecycle explained throughout the paper ensures that the answers returned a high quality (e.g., no duplicate instances, correct property values). An example of such a GDS can be found online[28].

---

[26] https://dialogflow.com
[27] https://rasa.ai
[28] https://www.oberoesterreich.at/ - Flo-Bot virtual assistant.

## 3    Discussion and Lessons Learned

In this section, we discuss our lessons learned from implementing the presented methodology, from the perspective of the overall approach and individual steps in the methodology.

### Community effort needed to maintain existing research products

There is a plethora of research that resulted in various tools for creation and curation processes. Unfortunately, many of them were abandoned in their GitHub repositories, and not maintained further. This is typically a consequence of research projects being over and researchers moving on to other projects. Community groups such as Knowledge Graph Construction (KGC CG) may be the solution to this "research prototype graveyard" situation. Such groups consisting of research and industrial partners can take a selection of approaches and tools and further maintain them as an open-source community effort.

### Real data is not perfect, knowledge creation is not trivial

Constructing Knowledge Graphs from heterogeneous sources scale well with declarative mappings[29]. There are many alternatives with different advantages and limitations. We decided on RML as it supports various syntaxes including a YAML-based one, with which the developers of our adopters more comfortable. RML also has its limitations. For example, frequently, we encountered data sources that do not provide any unique fields to join two logical sources (e.g., events and their organizers) but the relationship is specified by nested structures. We worked around this by extending the existing JSON-Path and XPath implementations with a ~PATH term which represents the absolute path of a value in the JSON or XML tree. These paths can be used to join different related logical sources without and fields to join. Here again community efforts like KGC CG can be beneficial for identifying common challenges in declarative mappings and addressing them within the existing tools and approaches.

### Quality assessment requires significant manual labor

One lesson we learned from the Knowledge Assessment process is that the level of automation varies a lot between Knowledge Graphs and there is no one-size-fits-all solution. The automation for the configuration and assessment may be increased if Knowledge Graphs provide more machine-understandable metadata such as VoID[30] descriptions.

---

[29] See here for a comprehensive list: https://stiinnsbruck.github.io/lkgt/
[30] https://www.w3.org/TR/void/

**There can be different perspectives on knowledge integrity**

One experience we had with our use cases is that the different instances of the same type may have different expected shapes. For instance, a generic Organization shape may require schema:vatID property however for an Organization instance that is the value of organizer property of an Event only the name property may be interesting. A SHACL shape that targets the Organization type would verify both Organization instances, which is not the intended behavior. To address this, we see domain-specific patterns as types with local properties and ranges. This means the relevant shape of each instance has to be asserted on that instance. Then the verification turns into instance checking under Closed-World Assumption.

**Necessary trade-offs in duplication detection**

As for many challenges in computer science, there is a trade-off between the run-time and the quality of the result. There are duplicates already lost in the pre-filtering phase which is a necessary evil to make the run-time feasible if the software shall run on a usual computer.

## 4    Conclusion and Future Work

Knowledge Graphs is a flexible and scalable solution for integrating large heterogeneous data. In this in-use experience paper, we summarized our work for building Knowledge Graphs from tourism-related data to empower various applications, especially conversational agents. Building Knowledge Graphs involve not only constructing them from different sources in different formats but also maintaining their quality in different dimensions.

In the scope of our work, we proposed a methodology, a lifecycle-based approach to construct and maintain Knowledge Graphs. We developed a set of tools to address the different tasks in the lifecycle. Although the majority of the tasks in the Knowledge Graph Lifecycle have been covered, there are still some tasks such as validating Knowledge Graphs against real-world, fusing linked instances, and automation of error correction that require further research. Moreover, the maturity of our tools is at a different stage, however, they are actively being developed by industrial adopters such as Onlim. We provided an evaluation of different tools supporting the lifecycle individually. Their real evaluation will be in the next couple of years as the developed approaches and tools are continuously being tested in the Knowledge Graphs and applications of Onlim. Alongside the development of individual tools, another feature challenge that will be addressed is the orchestration of all these tools within the lifecycle, which is when each task is supposed to run and how they are supposed to be harmonized.

Perhaps one of the biggest knowledge graph-building challenges our approach will tackle is the German Tourism Knowledge Graph. The presented methodology is currently being adopted to build a Knowledge Graph for German tourism,

which will integrate tourism data from the tourism marketing organizations of 16 federal states. The project started in December 2020 and the initial prototype will be ready in May 2021[31].

As a Knowledge Graph gets bigger and supports more applications, it may come to a point that the curation process may be infeasible, both due to the size of the Knowledge Graph and changing contexts (e.g., different applications and customers may have a different set of constraints and rules). Therefore, our further research will focus also on building a layer on top of Knowledge Graphs that enables applications to work on small subsets of Knowledge Graphs with different configurations for curation which will allow the customization of the Knowledge Graph for different application contexts.

## Acknowledgement

This work has been partially funded by the industrial research project Mind-Lab[32].

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
