# OpenReview forum: "Knowledge Graph Lifecycle: Building and maintaining Knowledge Graphs"
_eswc-conferences.org/ESWC/2021/Workshop/KGCW — KGCW 2021_

### Official Review · ~Jakub_Klímek1 · 2021-03-31
**Overview of tools applicable to the process of knowledge graph creation and curation, lacking conciseness and clear focus.**

**Rating:** 4
**Confidence:** 4

**Review:**

The paper describes a knowledge graph creation and curation process as used by Onlim, including the various tools used for its individual steps. Overall, the paper lacks a clear focus and conciseness, as the authors try to cover the whole process of KG creation and curation, which would deserve more space and a more detailed description of the individual steps.

Major issues:
1. p4, section 2.2.: It is unclear for what exactly MongoDB is used in addition to GraphDB. I suspect that by "individual annotations" the authors mean annotation extracted from a single web page, but I am not sure, and it is unclear how exactly it is challenging to retrieve the same data from a SPARQL endpoint, e.g. by utilizing the Graph per source named graph pattern.
2. The authors overuse footnotes - there are too many and sometimes there are whole paragraphs in them. It is unclear whether the reader should read their contents (then they should be in the main text of the paper) or not (then they can be omitted, saving space for more related work, the lack of which is mentioned in footnote 3 on page 2)
3. on page 8, the VeriGraph tool, described as SHACL-based is evaluated against ShExJava (a reference is missing). ShExJava is, however, implementation of ShEx - a different language. It is unclear how these two were compared, whether e.g. the used SHACL rules were rewritten to ShEx somehow.
4. In section 3, the authors talk about a presented methodology, however, the description of the KG creation and curation process in Section 2 is far from what could be called a methodology. It reads more as notes about the sets of tools usable for individual process steps. A methodology would need to be more formal.

Minor issues:
- p4: "knowledge on two different stores" => "in different stores"?
- p7: "with the caching mechanism" => "with a caching mechanism"?
- p12: "without and fields to join" => "any fields"?

---

### Official Review · ~Mario_Scrocca1 · 2021-04-13
**Discussion of tools lacking a proper framework**

**Rating:** 4
**Confidence:** 4

**Review:**

The paper describes a set of tools supporting a methodology for the knowledge graph lifecycle. The authors also present a preliminary evaluation of these tools and lessons learned from their in-use experience applying the methodology and the tools. Despite presenting tools covering relevant topics for the workshop and offering useful insights on their implementation, the paper lacks a proper framework and fails to support the claimed contributions, in particular:
- an overall description/discussion of the different steps of the methodology is missing at the beginning/before section 2 (it is difficult to grasp the overall methodology in the following subsections mainly presenting the details of the different tools)
- a structured discussion based on the in-use experience is missing, the different tools are mainly discussed and evaluated independently
- the lessons learned seem very decoupled insights, their description is not clearly bound neither to the different steps of the methodology neither to a specific use case

Minor issues
- A related work section/paragraph discussing at least the relation of this paper with works [5], [12] and [14] (mentioned when introducing the methodology, and also in Fig. 1) would help the reader
- Personally, I think the term "Knowledge Deployment" for the last step of the methodology is somehow misleading. The presented discussion is more related to the “exploitation" of the knowledge graph and not to the "deployment activities".
- Typos
    - Fig 2. Preperation -> Preparation
    - pag 7. "following the all outgoing edges" -> “following all the outgoing edges”
    - pag 12 in the 2nd lesson learned. "adopters more comfortable" -> “adopters are more comfortable”

---

### Official Review · ~Semih_Salihoglu1 · 2021-04-17
**The paper can generate interest because it discusses experiences from a real-world knowledge graph platform**

**Rating:** 6
**Confidence:** 4

**Review:**

This paper describes multiple steps of developing a knowledge graph-based application, referred to as knowledge graph lifecycle. This lifecycle is described from the perspective of products/services that Onlim, GmbH is offering. The steps include (can be seen in Figure 5) Knowledge Creation, Knowledge Hosting in a database management system, Knowledge Assessment, Enrichment and Cleaning, and Knowledge Deployment. I appreciated parts of the paper that gave references to specific technology the company's services are using (e.g., the use of PROV-O or RocketRML, or Apache Nifi). I also find value in learning about some deployed real-world applications (e.g., the Flo-Bot chat bot https://www.oberoesterreich.at/).

The paper can be improved significantly. The primary shortcoming is that because the authors cover many steps, they are very high-level and brief in each step. I think the part of the paper that has the most potential to generate interest is Section 3, "Discusssions and Lessons Learned". The lessons described here are not very interesting and vague. My suggestion to improve the paper is to: 1) scope down Section 2 to cover only 1 or 2 steps of the lifecycle but in more detail, possibly providing more technical description. For example one possibility to generate interest is to discuss alternative technologies the authors have considered and justifying more why they chose a particular technology (e.g., why MongoDB and not something else?). These decisions must have been made through some insights the authors had but the paper does not convey these. 2) The paper can be a lot more specific in Section 3 about the lessons.  For example, the community effort needed to maintain research projects: why is this important? Where there tools that the authors relied on from researchers but because the tool was no longer supported, they had to move away, so this wasted their time? Not enough context is given here.

Overall, although I think the paper has a lot of room for improvement, it is difficult to jump up and down for it. But I also think it can generate interest in the workshop, so I think the PC chairs should consider it for publication.

Minor Comments
constiute -> constitute
”extracting  -> quotation marks in the wrong direction (this should be changed throughout the paper)
platform .  -> remove space
with which the developers of our adopters more comfort- able. -> this sentence is missing a verb

---

### Official Review · ~Joshua_Shinavier1 · 2021-04-18
**Lacks focus and adequate support for key claims, but is an interesting case study**

**Rating:** 6
**Confidence:** 4

**Review:**

This paper contains an overview of a multifaceted knowledge graph construction project. It provides sufficient detail for the reader to understand the specific approach to KG construction and curation which was followed, and for that reason, may be of interest to others who are considering, or are already engaged in, a similar approach. The shortcoming of the paper is that it spends a great deal of time on the "what" and "how" of this approach, without exploring the "why", or comparing the approach to other alternatives which may have been considered. I think ideally, the paper would be organized around the "lessons learned" which are provided at the end. Certain key components of the paper, including the evaluation and the authors' claim that VeriGraph outperforms a number of other tools and frameworks, are not sufficiently demonstrated.

Ideally, the paper would be revised to address some of these shortcomings, though I think there is some value in including it in the workshop in its current form. Additional notes are provided below:

* Incomplete header: firstname.lastname@sti2.at etc.
* grammatical errors (e.g. missing definite article) and spelling errors (e.g. "preperation")
* "Knowledge Graphs integrate data from heterogeneous sources and can rapidly become quite large" -- what do the two statements have to do with each other? The second statement is also trivial: "They can be a useful resource to power many applications in different domains". I would just remove this fluff from the intro, including sentence #3.
* You mention "two main challenges" for a KG's lifecycle, but to my mind, the first challenge is actually two: how to integrate data sources as a graph, and how to make the graph scale.
* I would decapitalize "Knowledge Graph" as "knowledge graph" unless you are talking about Google
* Footnote #3 can be abbreviated to the last sentence. No need for a detailed explanation of why there is no related work section.
* Although a citation is provided to a more detailed description of the RML-based mappings, it would be interesting to see a brief example in the paper itself. Defining good mappings is one of the major challenges of KG construction from structured sources.
* In the Knowledge Cleaning subsection, it is not immediately clear in what way VeriGraph has been compared against the competing tools. AllegroGraph, for example, is simply a triple store, not a data quality framework. In the case of the others, how did you guarantee that this was an apples-to-apples comparison? VeriGraph's outputs are presumably very different from those of Stardog ICV, RDFUnit, etc.
* The Lessons Learned are good, and are perhaps the main contributions of the paper, given that much of the technical work is already described in greater detail in other papers. I would consider summarizing these lessons in the introduction.

---

### Meta-Review · Program_Chairs · 2021-04-21

**Recommendation:** Conditional accept
**Confidence:** 5

**Metareview:**

This is a paper that describes experiences from a large-scale knowledge graph construction project. It is an exemplary paper of what the organizers of the workshop consider as an experience paper and most reviewers agree. However, it is also clear from all reviews that there is still a lot of room for significant improvement. Thus, half of the reviewers doubt if it can be published in its current state while the other half argues that, despite the room for improvement, it would be of value if it gets accepted for presentation because it will spark interesting discussions. The organizers of the workshop lean to agree, as workshops are traditionally forums for discussion and exchange of ideas that allow the state of the art to advance. However, we would opt for a conditional accept, expecting that the authors will improve the paper before we agree that it will be accepted for publication and included in the proceedings. In more details,

* The paper lacks a clear focus and conciseness, as the authors try to cover the whole process of KG creation and curation. As such, they are very high-level and brief in each step. It would be better if the authors are **more detailed for 1 or 2 steps of the lifecycle**, possibly **providing more technical descriptions for these steps**.

* The reviewers argue that the authors talk about a methodology, but it is in fact not a methodology but rather a framework or something alike. We agree with the reviewers and we would suggest the authors to **explain better the contribution** which is not the methodology (if we properly understand the methodology is already published?), but the tools that are used to materialize the methodology.

* We also agree with the reviewers that the paper would benefit from an overall description/discussion of the different steps of the methodology. We suggest to follow the reviewers recommendation and **include a high-level description of the tools used to cover the methodology** before section 2 or at its beginning.

* The reviewers find value in learning about some deployed real-world applications but the paper spends a great deal of time on the "what" and "how" of this approach, without exploring the "why". We would suggest the authors to **be more detailed on their explanations of why certain choices were made**, **discuss alternative technologies** that were considered and justify more why they chose a particular technology. For instance, why MongoDB and not something else? what else was considered? then why MongoDB is used in addition to GraphDB?

* The reviewers agree that the paper fails to support claimed contributions. Certain key components of the paper, including the evaluation, are not sufficiently demonstrated. We would suggest the authors to **support their arguments with proper proofs** or tone down in the cases that a certain claim cannot be supported.

* The paper can be a lot more specific in Section 3 about the lessons which are currently vague and decoupled from the different steps of the methodology or the specific use case. We would suggest to the authors to summarise the lessons in the introduction, associate the lessons learnt with the rest of the paper, and include a structured discussion based on the overall in-use experience.

* We would also suggest to **address the more detailed comments of all the reviewers** and **fix all grammar and syntax errors**.

---

### Decision · Program_Chairs · 2021-04-23

Accept